# Genomic Approaches to Identify Molecular Bases of Crop Resistance to Diseases and to Develop Future Breeding Strategies

**DOI:** 10.3390/ijms22115423

**Published:** 2021-05-21

**Authors:** Antonia Mores, Grazia Maria Borrelli, Giovanni Laidò, Giuseppe Petruzzino, Nicola Pecchioni, Luca Giuseppe Maria Amoroso, Francesca Desiderio, Elisabetta Mazzucotelli, Anna Maria Mastrangelo, Daniela Marone

**Affiliations:** 1Council for Agricultural Research and Economics, Research Centre for Cereal and Industrial Crops, S.S. 673, Km 25,200, 71122 Foggia, Italy; antonia.mores@crea.gov.it (A.M.); graziamaria.borrelli@crea.gov.it (G.M.B.); giovanni.laido@crea.gov.it (G.L.); giuseppe.petruzzino@crea.gov.it (G.P.); nicola.pecchioni@crea.gov.it (N.P.); annamaria.mastrangelo@crea.gov.it (A.M.M.); 2IEEE, Institute of Electrical and Electronics Engineers, 22040 Anzano del Parco, Italy; luca.amoroso@ieee.org; 3Council for Agricultural Research and Economics, Genomics and Bioinformatics Research Center, Via San Protaso 302, 29017 Fiorenzuola d’Arda, Italy; francesca.desiderio@crea.gov.it (F.D.); elisabetta.mazzucotelli@crea.gov.it (E.M.)

**Keywords:** crop, disease resistance, genes, marker-assisted selection, meta-analysis, genomic selection, effectoromics, new breeding technologies

## Abstract

Plant diseases are responsible for substantial crop losses each year and affect food security and agricultural sustainability. The improvement of crop resistance to pathogens through breeding represents an environmentally sound method for managing disease and minimizing these losses. The challenge is to breed varieties with a stable and broad-spectrum resistance. Different approaches, from markers to recent genomic and ‘post-genomic era’ technologies, will be reviewed in order to contribute to a better understanding of the complexity of host–pathogen interactions and genes, including those with small phenotypic effects and mechanisms that underlie resistance. An efficient combination of these approaches is herein proposed as the basis to develop a successful breeding strategy to obtain resistant crop varieties that yield higher in increasing disease scenarios.

## 1. Introduction

Pathogens, represented by fungi, oomycetes, bacteria, and viruses, are some of the main limiting factors in agricultural quality and production [1]. The development of highly resistant varieties carrying long-lasting and broad-spectrum disease resistance is an economical and ecofriendly alternative to expensive and environmentally harmful chemical controls [2,3]. Niks et al. [4] distinguished different types of host resistance, depending on whether it refers to its effect on the phenotype or mode of inheritance. For the mode of inheritance, genetic resistance based on monogenic inheritance, often referred to as qualitative resistance, follows the gene-for-gene interaction model, provides a near-complete defense, but only against avirulent pathogen genotypes. Qualitative resistance, determined by major genes, is more suitable for manipulation in plant breeding, even if it has a limited duration because pathogens frequently adapt to and overcome genetic resistance. In contrast, quantitative genetic resistance controlled by several genes/QTL (Quantitative Trait Loci), shows complex multigenic inheritance, making breeding efforts challenging [5,6]. Traditional breeding programs have produced many significantly improved varieties over the past 100 years. However, progress was slow, in part due to lengthy of breeding cycles and the long time from cross to cultivar release [7]. Genes that contribute to pathogen tolerance/resistance can be obtained from local germplasm resources, or through exotic lines, wild species or genera, or lines from other breeding programs [8]. The development of molecular markers has created a powerful and practicable tool to perform gene selection in plant breeding. Today, in the post-genomic era, the availability of genomic tools and genetic resources is leading to development of new generation methods in plant breeding, which facilitate the study of the genotype and its relationship with the phenotype, in particular for complex traits.

This review reports a deep overview, the most comprehensive as possible, of the molecular approaches to find molecular markers, both genome-wide and closely linked to resistant genes, for genotypic characterization in crop breeding. Traditional and most recent strategies are herein described from Marker-Assisted Selection (MAS), Genomic Selection and Machine Learning to genetic engineering, as represented with a temporal and historical scale of their development in Figure 1. The practical applications of these approaches are discussed in light of their possible combination in breeding programs for improving plant disease resistance in crops.

## 2. Toward the Identification of Resistance Genes/Loci

To date, traditional mapping approaches have made possible the identification of genes/loci for disease resistance in many crops, useful for genetic improvement programs. Advanced genomic tools allowed us to accelerate gene identification—thanks to the growing availability of genome sequence data—and to determine a more precise location of the causal gene. Here, we reviewed mapping approaches from traditional to NGS-based ones, and meta-QTL analysis as tools to compare results from independent experiments and find consensus QTLs.

### 2.1. Traditional and NGS-Enabled Mapping Approaches

In crop plants, linkage analysis and genome-wide association studies (GWAS) have been extensively used to identify genomic loci responsible of resistance phenotypes [9]. A typical quantitative resistance locus (QRL) identified through linkage analysis encompasses hundreds of genes, and many credible candidate genes may exist among them, making it very difficult to identify the true causal gene. The identification of QTLs associated with agronomic traits in crops, and disease resistance above all, have been recently accelerated thanks to NGS-based technologies. Indeed, sequencing of crop genomes and transcriptomes have provided huge and comprehensive sequence data necessary to develop high density SNP arrays. Today, high-capacity SNP arrays are available for a broad range of plant species and are becoming widely used in the breeding of major crops like maize (50–600 k SNPs) [10,11], rice (51.5 k SNPs) [12,13], wheat (9 k, 35 k, 90 k, 660 K,420 K, 820 K) [14,15,16,17], potato (8.3 k SNPs) [18], barley (9 k SNPs) [19], soya bean (50 k SNPs) [20], rapeseed (60 k SNPs), [21] or sorghum (3 k and 90 k SNPs) [22,23]. At present, the Infinium platform from Illumina Inc. (San Diego, CA, USA) and the Axiom technology from Affymetrix Inc. (Santa Clara, CA, USA) are the most widely used platforms for large-scale SNP genotyping in crop plants [24]. As an alternative, genotyping based on sequencing (GBS) has been served as an application of NGS protocols for discovering and genotyping SNPs in crop genomes and populations, upon constructing reduced representation libraries of the target genomes as in GBS [25], specific locus amplified fragment sequencing (SLAF-seq) [26], or restriction site-associated sequencing (RADseq) [27]. GBS can be an effective option for orphan crops, while fixed genotyping chips are often preferred to GBS technologies for scenarios aiming to generate structured data sets of common sequence variants at low cost, with minimal bioinformatic input, for example within an ongoing breeding program [28]. On the other hand, to be effective, a fixed SNP genotyping platform must be applicable to a wide range of different genotypes; hence, the alleles of the chosen SNPs must be representative even for diverse germplasm, otherwise this could cause an ascertainment bias [29]. GBS-based genotyping methods can be more suitable for identifying true, causal genetic variants for phenotypes with a complex genetic architecture because these are typically influenced in crop species by rare alleles that may not be adequately represented on an SNP array [30]. Nevertheless, given the relatively high extent of linkage disequilibrium (LD) throughout the genomes of most crops, SNP markers on a fixed, high-density array are still likely to exhibit genetic associations with phenotypic variation through LD to the causal genes [31]. Moreover, for some species, different arrays have been designated to specifically address breeding or diverse germplasm, as in the case of wheat for which users can choose among the Axiom^®^ 35K Breeder Array and the Axiom^®^ Wheat Relatives Array.

In respect to linkage mapping, widely used from 30 years, genome-wide association mapping studies, firstly applied in human genetics, provide much higher resolution mapping and facilitate the identification of candidate genes for validation by transformation and/or mutagenesis [32,33]. Indeed, in many species/germplasm collections where LD decay is fast or genome size is limited, the resolution provided by GWAS supported by high SNP density is enough to decrease the number of candidates to few genes. For instance, a collection of Ethiopian sorghum of around 600 accessions genotyped by more than 200 thousand SNPs enabled the detection of a low frequency locus of about 78-kb region which includes three clustered R genes [34]. In other crop species, LD may often range over several hundred kilobases, especially in self-pollinating crops such as rice, soybean and wheat [35]. This results in the inclusion of many candidate genes in a single LD block exhibiting a significant signal, thus entailing the need for additional experiments to conclusively identify the causal genes. An efficient way to overcome this limitation is to construct haplotypes based on LD and then to perform haplotype-based association analysis. A SNP haplotype refers to a distinct combination of SNPs within LD blocks which tend to be inherited as an entire unit from a parent to its progeny. SNPs that can differentiate haplotypes are ‘haplotype tags’ and can be used as important genetic markers for MAS and genetic mapping. It is intuitively plausible that the information provided by SNPs is most useful when several closely spaced SNPs completely defining haplotypes in the target region are examined, particularly when the causal locus is unknown because they can provide greater QTL detection power and mapping accuracy than single markers. Indeed, examples have been reported about significant associations between haplotypes and phenotypes that were not detectable by a single SNP analysis as well as increase of phenotypic variation explained [33,36]. However, these advantages depend on models relating genotype to phenotype, that is the genetic architecture of the target traits, and demographic scenarios, and thus that power of QTL mapping with haplotypes must be evaluated on a case-by-case basis [37]. About disease resistance, haplotype-based association analysis has been deployed above all to characterize the diversity at single target locus in diverse germplasm [38], to accelerate the fine mapping of genomic regions containing known resistance loci [39], to validate target known loci in different genetic populations to develop robust and breeder-friendly markers [40,41], to characterize the track of allelic selection across the crop history in order to choose what type of alleles should be introduced into current cultivars [42,43]. In plant genomics, only few methods for haplotype-based GWAS have so far been proposed [44]. They require haplotype information a priori and therefore are difficult to establish at a genome-wide level. In perspective, this limitation can be overcome by means of the continue advancement of high-throughput sequencing technologies, which may enable the rapid and accurate resequencing of many genomes and therefore the definition of comprehensive haplotype maps. This progress is expected to revolutionize GWAS because haplotype maps will be a valuable tool for imputing genotypes and transferring sequence-level variation data across multiple gene mapping projects, thereby increasing the power and precision of trait mapping in GWAS and helping to understand better the basis of phenotypic traits [34,45].

### 2.2. NGS-Enabled Fine-Mapping/Cloning Approaches

A plethora of NGS-based mapping approaches have been developed so far and applied in the last decade to the mapping/fine mapping toward cloning disease resistance loci, as well as development of diagnostic markers for use in breeding. Some of them are suited for biparental segregating populations and integrated traditional bulk segregant analysis (BSA) with sequencing methods to map major resistance loci. They are named as QTL-Seq, Seq-BSA, or Indel-Seq when mainly focused on variations identified in insertions and deletions [46]. By allowing placement of QTL within a smaller genomic segment, QTL-seq facilitates both detection of QTL and its fine mapping at a stretch, thus allowing the rapid discovery of candidate genes for the trait of interest. Successfully deployed for the first time for faster identification of QTLs for blast resistance in rice (*Oryza sativa*) [47], QTL-seq has been exploited in many different species. Examples are reported in pigeonpea for Fusarium wilt and sterility mosaic disease resistance [48], in groundnut for bacterial wilt [49] and late leaf spot [48,49,50,51], in pepper for Cucumber mosaic virus [52], in tomato for leaf mold [53], and in watermelon for Fusarium wilt disease [54]. When QTL-seq is applied to several mapping populations derived from crosses with at least one common parent, it is named multiple QTL-seq (mQTL-seq) [55]. The utilization of multiple mapping populations representing a broader genetic diversity was beneficial for the validation of QTLs, along with narrowing down the detected QTLs to shorter segments for several agronomic trait. For example, mQTL-seq was applied on two sets of extreme *Ascochyta blight* (AB) phenotype bulks derived from Cicer intraspecific and interspecific crosses, allowing the physical mapping of three associated regions and identifying the AB responsive gene *CaAHL18* (*AT-HOOK MOTIF CONTAINING NUCLEAR LOCALIZED*) as a candidate for one of them [56]. In all these approaches, NGS is used to sequence bulks of contrasting phenotypes and to provide the assembly of the parent genome. For some crop species which have a large and complex genome, such an approach is cost ineffective and needs custom bioinformatic pipelines for genome assembly. A kind of genome reduction is therefore necessary to manage with these big genome crop species. Bulked segregant RNA Seq (BSR-Seq), which is based on whole transcriptome sequencing of contrasting bulks, represents an effective option [57]. Indeed, BSR-Seq approach is thus being widely adopted for rapid discovery of genes and markers highly linked with the target genes [58,59,60,61]. For instance, through BSR-Seq, the *Yr15* gene, which imparts resistance to yellow rust in wheat, has been fine mapped to a 0.77 cM region allowing the development of effective markers for MAS [62]. Specific-Locus Amplified Fragment (SLAF)-Seq is another approach to perform whole genome resequencing by simplifying genomes, but its successful application relies on a known reference genome and bioinformatics foundation [63]. Genes responsible for aphid resistance in soybeans have been identified using this approach [64]. It was also used to construct a high-density map with 2634 SNPs in watermelon with a considerably decreased distance between linked markers [65]. The combination between SLAF-seq and other genome-based sequencing methods can also lead to precise linkage mapping; for instance, SLAFseq and BSA-seq have been used to characterize the genes related to Phytophthora root rot in pepper [66,67].

When applied to genetic materials supporting high resolution mapping, NGS technologies have also accelerated the identification of the gene responsible for the target phenotype by forward genetic approaches. This is the case of NGS applied to mutant collections which generated a massive number of methods known as MutMap, MutMap Gap, MutMap+, modified MutMap, SHOREmap. The MutMap approach was proposed by Abe et al. [68] in rice to identify genomic regions governing important agronomic traits. In brief, this technique involves the generation of a mutant population using chemical mutagen followed by the selection of lines with desirable phenotype in M2 or subsequent generations. Then, similar to QTL-seq, whole genome sequencing is applied to contrasting phenotype bulks of F2 individuals generated by crossing of the selected mutant with the corresponding wild-type parent, grouped based on the target mutant phenotype. Modified versions of MutMap have been developed, as the MutMap-Gap, which facilitates identification of causal SNPs in genomic regions that are missing from the reference genome sequence [69]. Despite the number of methods, only few examples are available for disease resistance. For instance, MutMap was used in rice to clone the causative gene of a lesion mimic mutant, named lmm24, which exhibited enhanced resistance to blast fungus *Magnaporthe oryzae* and up-regulation of defense response genes [70], while the application of MutMap-Gap on the rice variety Hitomebore revealed the existence of the blast-resistance *Pii* gene, which was absent from the Nipponbare reference sequence [69].

More numerous applications have been reported for the MutChromSeq, which integrates chromosome sorting of selected mutants as strategy for genome complexity reduction before sequencing. Comparison of sequences from wild-type parental chromosomes with chromosomes from multiple independently derived mutants identifies causative mutations in a single candidate gene or a noncoding sequence. The power of this method depends on being a cloning by sequencing approach that is not relying on recombination-based genetic mapping and does not exclude any DNA sequence from being targeted. Because many R genes are present in gene families, with members in close physical proximity, the dissection of such loci by recombination is not practical and also further confounded by the extreme sequence diversity and R gene copy number variation often present between different haplotypes [71]. This approach was firstly applied to clone by sequencing the wheat dominant powdery mildew resistance gene 2 (*Pm2*) [71] and more recently in barley to clone the leaf rust resistance gene *Rph1* [72]. The disadvantage of MutChromSeq is that its application is limited to previously mapped loci. Similar to MutChromSeq, the TArgeted Chromosome-based Cloning via long-range Assembly (TACCA) approach combines lossless genome-complexity reduction via chromosome flow sorting with Chicago long-range linkage to assemble single chromosomes of complex genomes. Prior information about the mapped gene (flanking markers of a QTL) and its chromosomal location are used for chromosome sorting and sequencing. Thind et al. [73] cloned the leaf rust resistance gene *Lr22a* in wheat using this technique. After mapping the *Lr22a* in a 0.48 cM interval on chromosome 2D by two flanking SSR markers, the chromosome 2D was sorted and its de novo sequencing and assembly allowed the identification of the causative gene within four months.

NGS-based cloning approaches also aimed to harness the diversity of resistance genes within germplasm collections by deploying specific features of R genes. Indeed, most R genes encode proteins with nucleotide binding and leucine-rich repeats (NLRs), and plant genomes contain hundreds of NLR-encoding genes. Annotation of NLR-encoding genes in reference genomes has allowed the designing of probes to capture R gene fragments from a genomic library followed by sequencing. This method, called RenSeq, is therefore a cost-effective gene enrichment strategy which allows both the identification and the cloning of new R genes. As a proof-of-concept study, Jupe et al. [74] demonstrated its utility in potato and tomato. In this study, the target enrichment library was prepared using 523 NB-LRR-like sequences from potato genome, 57 tomato NB-NRC domains, and 9 characterized NB-LRR types from tomato, tobacco and pepper. Enriched samples from genomic DNA of the *S. tuberosum* Phureja clone were sequenced, and annotation was carried out resulting in a successful enrichment of NB-LRRs from 438 to 755. In a modified version called MutRenSeq, the comparison of the R gene complement of loss-of-resistance mutants with wild-type progenitors enabled the cloning of the wheat stem rust resistance genes *Sr22* and *Sr45* in two years only and without any positional fine mapping [75]. In a further option known as AgRenSeq [76], association analysis was combined with RenSeq to develop a method to identify R genes associated with resistance phenotypes of crop wild relatives, more in detail on the wheat relative *Aegilops* for stem rust resistance. The technique includes screening of wild plants for a variety of diseases, targeted resequencing of the wild plants to look for resistance genes, and association analysis of R gene variants with the phenotype scoring. Another RenSeq variant is represented by SMRT-RenSeq (single-molecule real-time RenSeq), employed by Witek et al. [77] to clone a gene responsible for resistance to *Phytophthora infestans* (*Rpiamr3i*) causing late blight disease in potato. This technology combines resistance (R) gene sequence capture (RenSeq) with single-molecule real-time (SMRT) sequencing. This approach should enable de novo assembly of complete nucleotide-binding, leucine-rich repeat receptor (NLR) genes, their regulatory elements and complex multi-NLR loci from uncharacterized germplasm [77].

Despite the number of NGS-based approaches developed so far, their application to disease resistance gene identification in crops is still too limited so far; therefore, further researches are needed in order to find new R genes/loci that can be then used in MAS or genome editing to improve cultivars.

### 2.3. Meta-QTL Analysis for Disease Resistance in Crops

Several requirements are needed to make MAS more efficient than phenotypic selection, including QTLs explaining a sufficient portion of the phenotypic variation, and precisely positioned onto the genome. A comparative analysis of the genomic regions responsible for the trait of interest across studies, genetic backgrounds and different environments could help to better understand the genetics of the trait, through the projection of QTLs on a consensus genetic map, or to the genome when available, and the identification of consensus QTL (meta-QTL) with a refined confidence interval (CI). When available, MQTLs can be anchored to a reference genome, and relevant candidate genes could be proposed. A number of studies since 2009 are reported in crops for meta-QTL analysis of resistance against fungi and viruses, in order to propose valuable QTL to perform MAS, in particular in *Theobroma cacao* [78], barley [79], tetraploid cotton [80,81], pea [82], wheat [83,84], maize [85], peanut [86], grapevine [87], and soybean [88]. A study carried out in maize on the response to stem borers and storage pests feeding on leaves, stems, and kernels, from geographically diverse environments, pointed out the presence of a total of 104 resistance meta-QTLs from 382 individual QTLs, many of which were involved in several insect species, therefore generating a significant interest for multiple-insect resistance breeding [89]. A deep analysis of the QTLome of Fusarium head blight (FHB) resistance, one of the most destructive diseases in wheat has been carried out by Venske et al. [90]. Their meta-analysis generated 65 meta-QTL from a total of 556 individual QTL found in the literature, distributed on all subgenomes and chromosomes of wheat. Candidate gene mining within the most refined meta-QTL on chromosome 3B, using the reference genome annotation, harvested 324 genes among which 10 were cross validated by publicly available transcriptional data as responsive to FHB. Two of these genes encode a Glycosiltransferase and a Cytochrome P450, previously verified as being responsible for FHB resistance and promising loci for breeding. The meta-QTL on 3B was centered with the *Fhb1* locus, known to have a large and stable effect on type II FHB resistance [91]. The same result has been reported in another study that summarized QTLs for FHB resistance, from five RIL populations, identifying six MQTLs with narrowed CIs compared to those of the original maps, as in the case of the 3DL MQTL from two original QTL with CI of 6.5 and 7.6 cM, respectively, that narrowed the CI to 1.6 cM [92]. KASP markers able to screen association mapping panels of elite lines and cultivars have been released, useful to transfer the linked MQTLs. Finally, partial resistance to white mold, a major disease that limits common bean production and quality worldwide, was investigated by meta-analysis [93]. Nine MQTLs have been identified from the 37 single QTLs for white mold resistance and anchored to the reference genome sequence, and candidate genes shown to express under *S. sclerotiorum* infection in other studies, including cell wall receptor kinase, ethylene responsive transcription factor, peroxidase, and MYB transcription factors, have been found within the confidence interval of five MQTLs. The recent availability of the sequence of complex genomes, as in the case of durum wheat, opened a new perspective in MQTL analysis through the projection of previously published QTLs directly onto the genome and the description of QTLomes for different traits, including resistance to diseases [94]. In conclusion, meta-QTL are generally mapped in smaller genomic intervals rather than original QTL, allowing identification of markers strictly associated and thus facilitating the search for genes underpinning the disease resistance.

## 3. Marker-Assisted Selection

Breeding activities are based on the collection and evaluation of genetic variation within a crop species, identification of superior alleles with a beneficial effect on the trait, promoting sexual recombination with elite genotypes, and selecting the individuals with the best phenotypic performance. In the last decades, selection assisted by molecular markers has moved selection from phenotype to DNA, with great advantages, depending on the trait, in terms of costs, time saving, and efficacy of selection [95]. Moreover, molecular selection can be done on seedling-stage materials, and it permits the enrichment of populations with heterozygous individuals by using codominant markers [96]. To some extent, QTLs, cloned genes, closely related markers, and donor lines have become available for different crops in the last decades, and they are actively transferred to elite lines through MAS. Many examples are available for improvement of resistance to rice blast, Magnaporthe oryzae, and bacterial blight, *Xanthomonas oryzae*, in rice, and many genes can be pyramided into the same genotype. Five resistance genes to bacterial blight (*Xa4*, *xa7*, *Xa21*, *xa13* and *Xa5*) contributed by IRBB66, and selected at the same time, allowed us to recover nine lines of Tainung82, one of the most popular japonica varieties, with a very high resistance level and good agronomic performance [97]. Other examples have been recently summarized for rice [98,99], chili [100], brassica [101] and wheat [102,103].

When the donor line is a wild relative of the crop, and this is often the case for disease resistance genes, linkage drag is a very common problem which can strongly limit the use of a particular gene in breeding. The size of the introgressed region—and therefore the probability that unfavorable alleles at physically linked loci can be transferred together with the gene of interest—can be reduced by background selection, made with a suitable number of markers to monitor the percentage of the recurrent parent genome. As an example, Baliyan et al. [104] introgressed *Xa21*, *xa13* and *Xa5* for resistance to rice bacterial blight from IRBB-60 to the widespread basmati cultivar CSR-30, through foreground selection for the three genes, associated with background selection to reduce as much as possible the percentage of the donor parent and to reduce the linkage drag. Nevertheless, when recombination is strongly suppressed in the cross between the elite cultivar and the wild relative, background selection is not enough in controlling this aspect. Due to linkage drag, the gene for resistance to fusarium head blight identified in an accession of *Thinopyrum ponticum*, *Fhb7*, has been transferred to common wheat, but not used in breeding. Crossing the donor line KS24-2 with the *ph1* mutant of Chinese Spring, a mutant in which homoeologous pairing and recombination are not suppressed, lines in which the size of the introgressed region was reduced to 16–17% of the original one were obtained [105].

One of the advantages offered by MAS compared to the phenotypic selection is the shorter time requested to obtain improved lines, thanks to the rapidity of the molecular assays and to the possibility of testing lines in their juvenile growth stages. In this regard, the length of the growth cycle of the crops still represents the limiting step in the process, in particular for tree species, and it is therefore an aspect that can be technically improved to make the whole process much faster. A method called “speed breeding” has been recently developed, which can be used to achieve up to five generations per year for spring wheat, durum wheat (*T. durum*), barley (*Hordeum vulgare*), chickpea (*Cicer arietinum*) and pea (*Pisum sativum*), and four generations for canola (*Brassica napus*), instead of two under normal glasshouse conditions. This result is achieved through the modulation of growing conditions in glasshouses and with the use of supplemental LED (light-emitting Diode) lighting [106]. Speed breeding can accelerate the breeding process, not only increasing the number of generations per year but also accelerating plant development for research purposes, including phenotypic evaluation of traits expressed in adult plants, mutant studies and genetic transformation. The length of the vegetative phase is an important factor limiting breeding in tree species. A method called high-speed breeding or fast-track breeding has been employed to reduce the duration of the juvenile phase prior to flowering and fructification in trees. This method combines genetic transformation and MAS in the same breeding program. Flachowsky et al. [107] obtained BC1 apple lines in which genes for resistance to apple scab and powdery mildew were pyramided through an MAS scheme in which one of the parents was a transgenic early flowering plant overexpressing a FRUITFULL homolog (*BpMADS4*), then eliminated by segregation. A similar approach has been developed using the integration of precocious transgenic trifoliate orange with the overexpression of *CiFT* (Citrus FLOWERING LOCUS T), applied to incorporate resistance to citrus tristeza virus of trifoliate orange into citrus germplasm [108]. Using fast-track breeding system, one generation of backcrossed breeding that would normally take at least five years was achieved in a single year.

Numerous studies pointed at comparing MAS with other selection criteria. Different results came out, but in general the choice of the best selection method depends on several factors, above all the nature and genetic basis of the trait for which the selection is applied. Castro et al. [109] recommended the use of MAS for four QTLs for resistance to ascochyta blight in chickpea because the use of markers reduced the time taken to select resistant lines even if it was more expensive than phenotypic selection. In another study, genomic selection resulted in higher selection efficiency than MAS for six traits related to reaction to fusarium head blight in common winter wheat, probably due to the complex quantitative genetic basis of resistance to fusarium head blight in this species [110]. Therefore, as indicated by the recent scientific literature, MAS can provide the best results when combined with other approaches in breeding programs, depending on the genetic control of the target trait. As an example, it has been applied to hybrids production in rice, in which selection for resistance genes is integrated with selection for hybrid genes related to male sterility [111,112,113]. In particular, Kim et al. [114] described a large breeding program in which hybrid rice lines highly resistant to diseases were selected based on three major hybrid genes, six genes for resistance to bacterial blight, four genes for resistance to blast, and other two genes for resistance to brown planthopper and tungrovirus.

## 4. Genomic Selection and Machine Learning

Plant breeding programs are typically time- and cost-intensive and, depending on the crop, it can take many years until a new variety is released. This reflects the fact that breeders, after creating new variability through the crossing and obtaining genetically stable lines, have to test variety candidates in multilocation trials in order to select superior genotypes with a high agronomic performance across a range of different environment conditions at the highest possible precision. In the pre-genomic era, common breeding strategies involved the use of classical quantitative genetic approaches, including pedigree information to estimates Best Linear Unbiased Predictors (BLUPs) [115]. Since the 1990s, advances in molecular genetics techniques has led to development of different molecular marker systems which drastically increased the total number of polymorphic markers available to plant breeders, revealing widespread genetic variation in genomes. More than 10,000 QTLs using different marker systems have been reported in more than 120 studies covering 12 plant species that aimed to improve quantitative traits of economic importance [116]. In the last two to three decades, molecular markers have been integrated in the conventional breeding by applying MAS for single trait provides selecting individuals with QTL-associated markers that have major effects. However, agronomically important quantitative traits are often controlled by many small-effect genes, which have been difficult to take advantage of in practical breeding [117]. Indeed, small-effect loci are difficult to map, and, whenever mapping is successful, often multiple QTLs are present, which are difficult to use simultaneously in breeding. Consequently, MAS had limited success in improving quantitative traits. Moreover, attempts to improve complex quantitative traits by using MAS have been unsuccessful due to the instability of QTL across multiple environments, due to QTL x environment interactions [118]. More recently, the possibility of using high-density SNP arrays allowed the development of statistical models to predict marker trait associations accurately, depending on the genetic architecture of the predicted trait. One of the most widely used strategies involves using the additive relationship matrix estimated from markers instead of the additive relationship matrix estimated from pedigree with BLUP models. This was the beginning of the genomic (or genome-wide) selection (GS) era, and the new BLUP model was called a G-BLUP model [119,120]. The GS is a method that has promised to overcome the limitations of MAS for quantitative traits [121], with the objective to determine the genetic potential of an individual instead of identifying the specific QTL. Over the past two decades, several different statistical models have been proposed for GS, including methods which assume the following: (i) a normal distribution of SNP effects (e.g., Genomic BLUP—GBLUP, Ridge Regression best linear unbiased prediction—RR-BLUP); (ii) a prior distribution of effects with a higher probability of moderate to large effects (BayesA, weighted Bayesian shrinkage regression wBSR); (iii) some SNP effects are zero (BayesB, BayesCπ); (iv) nonparametric methods (random forest, reproducing kernel Hilbert space (RKHS) or neural network approaches) [122]. To implement GS, a training population and a test population are first established, which consist of individuals included in the reference population, with phenotypes for the target trait(s) and genome-wide DNA marker genotypic data [119]. The genotype and phenotype information from the training set is used to derive an equation that predicts the effect of each marker on the trait, with all marker effects fitted simultaneously. If the markers are in LD with the causal mutations affecting the trait, they will capture a large proportion of their genetic variance. Marker effects are estimated using individuals of the training population with both genotypic and phenotypic information. These effects are combined with marker information of an individual to calculate the genomic estimated breeding value (GEBV). To optimize the model, its predictive ability is calculated based on a cross-validation (CV) system using the training and the test population. Subsequently, GEBVs are estimated for the test population, and the predictive ability of the model is then calculated as the correlation between GEBV and phenotypes of the test population. For all major crop species, numerous studies have reported the successful selection decisions based on GS to improve accuracy of selection and speed of genetic improvement [123]. From these, at least three key learnings of practical importance from implementing GS in crop breeding programs can be derived. The first two concern the training population that must be very large and should include individuals (lines/varieties) closely related to the selection candidates [124]. Estimates of the number of loci affecting quantitative traits likely range from 2000 to 4000 [125]. Setting-up of the first training data is often a large investment, and breeders should consider that an increase in prediction accuracy is observed with increasing training population size, as reported for wheat for which a plateau at ~700 lines consisting of full-sib, half-sib, and less related wheat lines from three consecutive breeding cycles have been considered [126]. The third teaching concerns the reference population, which includes training and test populations, that must be frequently updated with new genotyped and phenotyped individuals to ensure the accuracy of the GEBV is maintained over time [127]. When quantitative resistance is based on many minor genes, GS should be preferred to MAS. Over the past several years, a number of studies demonstrated the effectiveness of the current GS models to capture and predict the genetic variation for disease resistance, particularly quantitative disease resistance, for example wheat rusts [128,129,130,131], Fusarium Head Blight in wheat [110,132], and Leaf Blight in maize [133]. Moreover, MAS could also be used together with GS in a breeding program to fix major QTLs in the F2 and F3 generations, followed by GS for resistance traits with a more complex genetic basis, in order to avoid useless evaluation of lines that do not carry essential QTL alleles [134] (Figure 2).

With advances in GS, volumes of data have dramatically increased, and consequentially also research efforts aimed at integrating and unifying several fields of research have increased, such as computer science or machine learning (ML). ML is a subfield of artificial intelligence (AI) that enables machines to improve at a given task with experience. There are three main branches of ML: supervised (models are trained using dataset with known features, learning to predict features for new and unseen elements of dataset), unsupervised (models are trained using nonlabeled data, they attempt to discover hidden patterns on their own and draw conclusions), and reinforcement learning (models learn over time by interacting with their environment, gaining rewards for successful actions and trying to maximize cumulative reward). The major difference between ML and statistics is their purpose. Statistical models are designed for inference about the relationships between variables, which is achieved through the creation and fitting of a project-specific probability model. ML models are designed to make the most accurate predictions possible, by using general-purpose learning algorithms, automatically tuning their parameters in the training phase to find the optimal patterns in often rich and unwieldy data. In other words, while statistical models try to find correlations in a dataset in order to relate data to output results and possibly predict future values, the goal of ML is to have predictions as correct as possible, not to find correlations; obviously it will start from a mathematical basis (statistical learning), evaluating all the possible functions that describe those data and choosing the one that provides the “minimum expected risk”. For example, in the analysis of genotypes, classical statistics analyses each pair of markers in association with the phenotypes independently of the other markers, to understand which ones are involved in the expression of a certain trait. By changing the focus from searching for associations to predicting, through ML the multivariate relationship between markers is validated by the prediction success on the dataset test: ML methods can identify complex interactions between attributes without making assumptions about the data to be analyzed, or at least make much more generalized assumptions, and thus simplifying the methods. As shown by Grinberg et al. [135] on three experiments under different conditions and on different datasets (yeast, rice and wheat), using and comparing the results of different statistical and ML models, ML models were better able to interpret the system of genotype/phenotype interactions. ML methods that currently prove to be most successful, which fall within the supervised learning, are SVM (support vector machine) and random forest, as also observed by Gonzalez-Camacho et al. [136] in the analysis of rust resistance in wheat.

A special case of ML is represented by artificial neural networks (ANN), which are computational models built on the brain functioning of vertebrates and most invertebrate animals. In particular, the behavior of biological neurons is replicated with mathematical functions; these neurons are linked together in a network by connections called synapses. The brain is not static but modifies itself, at different levels, based on incoming stimuli, physiological alterations, experience, learning, etc. Changes in synaptic forces (synaptic efficacy), the number and structure of the synapses themselves, define the synaptic plasticity [137,138] that is believed to contribute to learning and memory. In the simplest architecture of an ANN, one hidden layer, containing neurons, is present between the input and output. When many hidden layers are considered, we talk about deep learning (DL). To clarify the concept, we report a concrete example. Let us take many pictures of different leaves attacked by two distinct diseases that we would like our network to recognize. We supply the images as input and we read the output value, comparing it with the real value we expect (in this case we are talking about supervised learning because in the learning phase, we communicate the correct values to the network). If the output value is wrong, the learning algorithm will modify the weight values of the mathematical functions (neurons), trying to minimize a specific error function. By repeating this process several times, the network can come to recognize the incoming sample with higher probability. The effectiveness of this learning lies in the ability of the neural network, once properly trained, to generalize; that is, it will be able to classify the two types of diseases, even providing, as input, images that it has never processed. Simplifying, we can think that each hidden layer learns a certain characteristic of the input data. It will do everything by itself in the learning phase. After the learning phase the network is ready to predict new and never seen data; in fact, providing an input, the network will recognize some specific characteristics present in it, selecting them among the many it has learned, thus allowing the classification. The big advantage of such a model is that we don’t have to know a priori which characteristics the net will learn, and which will be useful for the classification. We will have a kind of black box able to classify data autonomously (Figure 2). It should also be emphasized that everything will be independent of the type of data provided. The network will learn to recognize the key characteristics for classification, based on the examples provided only. A binary classifier, like the example, is just one of the possible applications obtainable with DL. With different types of layers and architectures, other DL networks can perform various and more difficult tasks. Examples of practical applications of DL retrieved in literature include understanding the correlation between genotype and phenotype [139], or prediction of molecular phenotypes ab initio from a DNA sequence allowing to discover causal variants for genetic diseases [140]. In this case, the big advantage of DL networks, analyzing the genetic variants in the population, is just that only few variants will be used in the training phase, but predictions will be valid for all other variants [141]. Deep learning approaches were applied also in the phenotypic data analysis. Indeed, traditional methods of phenotyping have not kept pace with the available high-throughput genotyping tools, and researchers and engineers had to adapt newer technologies in field phenotyping to overcome this bottleneck. Singh et al. [142] reviewed examples of DL in plant biotic stress image-based phenotyping in different crops (e.g., in apple for Alternaria leaf spot and mosaic rust; in Cassava for brown streak disease; in tomato for yellow leaf curl virus; in grape for leaf blight; in wheat for powdery mildew). A study aiming at the identification of the Northern corn leaf blight in maize reported DL successfully used together with images from UAV (unmanned aerial vehicle) reconnaissance for the identification of the disease [143]. The advances in high-throughput phenotyping data allow to reinforce the concept of “envirotyping” that involves collecting environmental factors through multi-environment trials, geographic and soil information systems and empirical evaluations and has various applications, including environmental characterization, genotype x environment interaction analysis and phenotypic prediction [144].

## 5. Effectoromics

Among “omics” technologies that have provided new insights into the understanding and managing of pathogenesis, effectoromics is a novel approach that allows the identification of candidate pathogen effectors (*Avr*) and their use as functional markers to define corresponding host resistance (R) and susceptibility (S) genes. A review has been recently released on the evolution of methods used in effector discovery, from physical isolation and in silico predictions to functional characterization of the effectors of plant pathogens and identification of their host targets [145]. Few examples are so far reported in the literature on crops. For instance, for the tomato late blight, caused by *Phytophthora infestans*, three genes for field partial resistance (*Rpi-blb1*, *Rpi-blb2* and *Rpi-vnt1.1*), as well as the corresponding avirulence genes (*Avrblb1*, *Avrblb2* and *Avrvnt1*, respectively), have been discovered [146,147,148,149,150,151]. More recently, a new family of Phytophthora small extracellular cysteine-rich proteins (PcF/SCR) has been reported as recognized by solanaceous plants that, combined with the well-known NLR genes, might provide a tool to target a wide spectrum of the P. infestans population and contribute to potato breeding [152]. Several candidate effectors from wheat pathogens have also been identified, and some were functionally validated [153,154] and cloned [155]. In particular, susceptibility to tan spot caused by the necrotrophic pathogen *Pyrenophora tritici-repentis* is strongly correlated with plant sensitivity to the effector *ToxA* which triggers strong necrosis in wheat genotypes that carry the susceptible allele at the *Tsn1* resistance locus. A high-throughput screening procedure for evaluating wheat genotypes through the infiltration of *ToxA* into wheat leaves has been developed to identify commercial cultivars carrying the susceptible allele at the Tsn1 locus [156]. Nevertheless, in the case of diseases such as FHB or *Blumeria graminis* f. sp. *hordei*, the quantitative resistance is determined by a high number of effectors involved, and more efforts must be done to discover *F. graminearum* effector reservoir [157,158]. In particular, nearly 7% of the genome of the barley powdery mildew fungus encodes secreted effector proteins [158]. The development of genomics pipelines to populate sequence database of pathogens with repertoire of effectors, and corresponding expression data following plant pathogen interaction, would provide the necessary knowledge for the full deployment of this alternative approach for plant screening. For example, the Pathogen–Host Interactions database (PHI-base, www.phi-base.org, accessed on September 2019) is a manually curated database comprising over 6780 genes from 268 pathogens of over 210 hosts (September 2019), of which 60% are plants [159]. Thereafter, full deployment will be possible by combining effectoromics with targeted genome engineering approaches (genetic transformation, genome editing) of host plants.

Because poor information related to the study of effectors is available in crops to date, to the best of our knowledge, this highlights the need for further researches. We suppose that the availability of databases of effectors and the knowledge of crop and pathogen genome sequences could make modern approaches as machine/deep learning possible to predict genes that interact with them to confer resistance or susceptibility.

## 6. New Breeding Technologies (NBTs)

Good opportunities in improving pathogen resistance are offered by genetic engineering. Initially, most attempts had involved a transgenic approach in many crops [103,160,161,162,163,164,165,166,167] and some genetically engineered food crops have been approved for commercial production. This strategy followed by cross and MAS has also been applied to accumulate more genes/alleles in transgenic plants for reaching a durable resistance in crops, but it has many legislative and sustainability problems. More recently, the direct modification of the plant genome is possible through the insertion of the native resistant allele using genetic transformation, or via gene modification/editing approaches by using ecofriendly new breeding technologies (NBTs). The greater knowledge of the gene sequence and its precise position on the chromosome provided by advances of genomic and molecular techniques, as discussed in this review, and by the tools of bioinformatics contributed to this outcome. Among these, cisgenesis and intragenesis, and genome editing (GE) techniques have had the widest application. Some applications of RNA interference and transgrafting to improve disease resistance in woody fruit species have also been reported [168].

### 6.1. Cisgenesis/Intragenesis

The exploitation of the genetic pool of a species or of a closely related species, which is similar to that available for traditional breeding, can be applied to genetic modifications of crop plants by using cisgenesis or intragenesis [169,170,171]. The result is either the transfer of resistance genes from related species, or overexpression of those already present within the crop itself, with significant time savings and avoiding linkage drag in respect to traditional gene transfer obtained by crossing. Assumption of this approach is isolation of complete functional genes, together with their associated promoter/terminator (regulatory sequences), which is facilitated by the advancement of sequencing technologies and the availability of wide genome information. Unfortunately, this approach is limited to few crop species. In cereal crops, the use of cisgenesis in improving the pathogen resistance is currently limited to wheat [172], whereas horticultural, fruits and ornamental crops, many of which are heterozygous and vegetatively propagated, were successfully improved for disease resistance through cis/intragenesis approaches [173].

In potato, a Durable Resistance against Phytophthora (DuRPh) program was developed with the aim of introducing, by a cisgenic approach, several (RERR)-genes from wild potatoes, with their native regulatory sequences, or a stack of multiple R-genes to ensure a stable and widely efficient resistance, into cultivated susceptible cultivars and to verify their performance by field trials [171,174,175]. Moreover, by a marker-free *Agrobacterium*-mediated transformation, cisgenic potato plants by stacking blight resistance R-genes from *Solanum stoloniferum* (*Rpi-sto1*) and *S. venturi* (*Rpi-vnt1.1*) were also obtained, which showed broad-spectrum resistance to late blight without affecting the characteristics of the original variety [176]. Other examples of efficient application of these approaches are cisgenic melon plants developed by transferring the genes *At1* and *At2* from wild melon (*Cucumis melo*) to a susceptible variety that showed enhanced activity of glyoxylate aminotransferase and increased resistance against powdery mildew [177], and intragenic strawberry plants with overexpression of endogenous PG1P-RRRP under the control of the strawberry promoter from the fruit specific expansin gene (*Exp2*), very active in red, ripe fruits, which showed resistance to gray mold [178]. Cis/intragenesis has been effectively used for different woody fruit species including apples, utilizing resistance genes from wild Malus accessions to confer resistance against the fire blight and scab disease [179,180,181,182,183,184,185,186]. In grapes, the most important table variety in the world “Thompson Seedless” was engineered with a VVTL-1RRRP (*vvtl-1*) cisgene to constitutively express VVTL-1 (*Vitis vinifera* thaumatin-like protein), resulting in two lines with foliar delay in powdery mildew development and decreased severity of black rot, and berries with significantly lower sour-bunch rot incidence during field tests [187]. Finally, a “foreign DNA-free” intragenic vector has been developed in Citrus spp. cultivar (*Citrus paradisi*) by using C. clementina-derived T-DNA-like region [188].

Despite the greater safety of the genotypes obtained by cis/intragenesis, many measures must be adopted to improve the application of this approach, overcoming specific problems, such as the variability of gene expression or the silencing of endogenous genes depending on the position of insertion of the cis/intragene or the presence of extraneous sequences [171,176,185,189]. Neverthess, cisgenic lines could be a good starting point in the choice of parents to be used in a breeding program for disease resistance (Figure 2).

### 6.2. Genome Editing (GE)

Genome editing (GE) refers to an advanced genetic modification tool in which the genome sequences are edited with high efficiency and accuracy by using engineered nucleases, such as zinc finger nucleases (ZFNs), transcription activators like effector nucleases (TALENs), and, recently, clustered regularly interspaced short palindromic repeat (CRISPR) systems with associated protein 9 (Cas9) (CRISPR/Cas9) [190,191,192]. The nucleases, introducing double-strand breaks (DSBs) into a definite region of the genome, can precisely delete, replace, or insert specific sequences in a targeted site. The DNA repair machinery makes the modifications useful for gene knockout/in and to create new better alleles for a specific trait that can replace the unfavorable ones. The use of targeted nucleases to change specific genes is dependent on knowledge of the genomic sequences, which are now available for major crop plants.

Unlike ZFNs and TALENs, which use protein motifs for target identification, in the CRISPR/Cas9 system the specificity of the cleavage is governed by base complementarity between the CRISPR RNA and the target DNA or RNA molecules. It has many more advantages than the former, due to its simplicity, high efficiency, high specificity, easiness of use in laboratory, wide applicability, the possibility of multiplexing that simultaneously causes targeted mutations in multiple genes [193,194]. It is also not so expensive, and vector designing is relatively less complicated due to the availability and easy access to the improved bioinformatics tools [195]. Furthermore, designing two or more targets for one gene can improve the probability of obtaining homozygous mutations in the first generation [196]. This is particularly important for woody plants which have long generation times. Generally, potentially useful CRISPR/Cas9 mutations are not restricted to ORF regions because they can regard some cisregulatory elements, such as promoters [197] (Appendix A). The CRISPR/Cas9 system can achieve efficient and transgene-free editing in plants through different ways, i.e., Agrobacterium tumefaciens transformation, protoplast transformation or direct bombardment of guide RNA (gRNA) and Cas9 to plant cells [198,199]. Most of the modifications aimed to increase pathogen resistance were performed by using *Agrobacterium*-mediated transformation of leaves [200,201,202,203,204,205,206,207], cut cotyledons [208,209], epicotyls [207,210,211,212], immature embryos [213], embryogenic cells [214], floral explants [215], suspension cells [216], or protoplasts [217,218]. Moreover, the identification of genotypes with good transformation/regeneration performance, the standardization of transformation protocols and the increasing transformation efficiencies may facilitate the broad application of GE strategies [219,220]. GE has been demonstrated to confer resistance against major pathogens, including bacteria, fungi and DNA and RNA viruses (review in Appendix A). Regarding bacterial pathogens, many examples are available in which both S genes and negative regulators of plant defense response were edited target sites for resistance to rice bacterial blight caused by *Xanthomonas oryzaepv. oryzae* (*Xoo*). This is an important disease that depends on TALE-mediated induction of at least one member of the SWEET family of sugar-transporter genes involved in the efflux of sugar across the plasma membrane [221]. Some independent studies have applied TALEN and CRISPR/Cas9 to target either the effector binding elements (EBEs) within the promoter or the CDS region of some SWEET genes, also through a multiplex targeting approach, thus inducing a significant resistance against Xoo [214,222,223,224,225]. In tomato, as well as in citrus, increased broad-spectrum resistance against multiple plant pathogens was reached with knockout mutants of DOWNY MILDEW RESISTANCE 6 (*DMR6*), a gene encoding for a superfamily of 2-oxoglutarate (2OG) Fe(II)-dependent oxygenases involved in hydroxylation or desaturation steps in plant hormone synthetic pathways, specifically up-regulated during pathogen infection [209,226]. In both cases, an increase of salicylic acid levels and no alteration of plant development and morphology were detected. About the horticultural species, an improved resistance against tomato bacterial speck disease was also attained targeting mutation in *SlJAZ2* gene [227]. Recent applications of CRISPR/Cas9 in woody fruit species concerned the increase of resistance against Citrus canker caused by *Xanthomonas citri* subsp. *citri* (Xcc) through knock out of the transcription factor LATERAL ORGAN BOUNDARIES 1 (*CsLOB1*), that plays a critical role in promoting pathogen growth and erumpent pustule formation [210,211], or the resistance to fire blight disease caused by *Erwinia amylovora* in apple [218].

Regarding fungal pathogens (Appendix A), the well-known example of mutation in host S genes was represented by the Mildew resistance locus O (*Mlo*). Mlo encodes a membrane-associated protein involved in negative regulation of vesicle-associated and actin-dependent defense pathways at the site of pathogen penetration, and it is required for PM fungal penetration of host epidermal cells. It has been mutated to confer resistance to powdery mildew (PM) fungus *Blumeria graminis* f. sp. *tritici* in wheat [228,229]. In particular, Wang et al. [229] used TALENs and CRISPR-Cas9 to introduce site-specific mutations in a conserved region of the MLO exon 2 of all three homoeoalleles at the mildew resistance locus O (*MLO*) of wheat embryos using particle bombardment, and small deletions in the MLO locus were reported in all three genomes of primary transformants (T0), conferring broad-spectrum resistance to PM in homozygous mlo plants. Mlo gene was also edited in a transgene-free manner in tomato [230]. Among fruit species, in grapes, Wang et al. [215] targeted the VvWRKY52 transcription factor which has been shown to play roles in biotic stress responses, and homozygous mutants obtained in the first generation showed increased resistance to *Botrytis cinerea*.

Genome editing has also been applied to improve resistance against oomycetes. A study reported the editing of the effector gene *Avr4/6* in *Phytophthora sojae* that prevented its recognition by the corresponding soybean R proteins Rps4 and Rps6, demonstrating its involvement in immunity activation [231]. Moreover, GE provides an efficient tool against both DNA and RNA plant viruses (Appendix A). Designing sgRNAs to target viral genetic elements, MP (Movement Protein), IR (Intergenic Region), CP (Coat Protein), Rep (Replication association protein), LIR (Long Intergenic Region), prevents replication of viral genes and allows us to the development of resistance against viral pathogens by blocking its access to replication protein or causing error prone mutation of viral genome [202,203,204,232,233,234]. Recently, an endogenous banana streak virus (eBSV), a plant pathogenic double-stranded DNA virus, has been inactivated using multiplexing CRISPR/Cas9 system, and asymptomatic banana plants were obtained [216]. Cas protein variants from other bacterial strains, such as the Cas9 from *Francisella novicida* (FnCas9) and the Cas13a from *Leptotrichia shahii* (LshCas13a) or *Leptotrichia wadei* (LwaCas13a), were more effective against RNA viruses [206,235]. In particular, FnCas9 operated by RNA binding and not for cleavage capacity while the cleavage sites of LshCas13 were essential against different types of RNA viruses. For example, FnCas9 and its sgRNA were engineered to target CMV (*Cucumber Mosaic Virus*) and TMV (*Tobacco Mosaic Virus*) [236], and the LshCas13a system was successfully used to inhibit potyvirus infection in tobacco and potato [206].

Another useful approach to limiting the spread of the viruses and the evolution of the disease it is to identify host factors necessary to assist viruses in the various phases of infection [208,213,237]. As an example, targeting eIF4E via CRISPR/Cas9 system enhances1 resistance against many RNA based viruses, including Ipomovirus and Potyviruses in cucumber [208] and in rice [213].

There are some problems in the application of these strategies that must be taken into account, such as the possibility of impairing the growth of the crops, the risk of off-target effects or the evolution of mutations in targeted viruses, especially DNA viruses, to escape from CRISPR/Cas9 cleavage [238], and more researches are needed to improve these systems and avoid complications in addition to exploring all their potentialities. The choice of genomic targets essential for the replication or movement of the viral pathogen, multiplexing the guide RNAs to improve their robustness, or the use of Cas12a (also known as Cpf1) may reduce the occurrence of escape viral variants because mutations caused by CRISPR-Cas12a are less likely to abolish the recognition of the target by the original guide RNA minimizes viral evasions [204]. Edited plants in R genes obtained with the approaches here described could be useful to be integrated in crosses within breeding programs aiming to shorten the obtaining of lines with favorable allelic combinations and improved resistance, as proposed in the Figure 2.

## 7. Conclusions and Prospects for New Breeding Scenarios

Severity and frequency of disease occurrence are rising in light of changes in global climate, affecting crop production worldwide. With the need to accelerate the development of improved varieties, genomics-assisted breeding is becoming an important tool in breeding programs. An efficient combination of approaches based on DNA-markers, genomic sequence information and high-throughput phenotyping can help in obtaining improved cultivars that yield higher in an increasing disease scenario. A possible advanced breeding scheme to obtain genotypes with enhanced resistance, almost for herbaceous plants, as proposed in Figure 2, begins with selection of parental lines characterized by high level of disease resistance, due to major known resistance loci, but also to minor undefined alleles. Lines obtained from genome editing in R genes could be a good starting choice in a pyramiding gene program. Possibly, potential candidate lines could be also derived from screening for effector targets of resistance genes; those with a positive response to the effector are useful for breeding superior varieties. Molecular markers associated to alleles at major loci or perfect markers designed on the gene sequence in the case of a cloned R gene can be used in MAS in order to fix them and obtain the desired allele combination at the very first generations of the breeding scheme. Then, some genomic selection cycles can be carried out to select for minor quantitative resistance genes which are responsible for more durable resistance but are very difficult to identify and select for. Machine learning for big data management can also be integrated in this proposed breeding scheme which complements MAS and genomic selection, in particular supporting phenotypic data acquisition and interpretation and breeding value prediction models. Such an integration of different approaches can strongly contribute to shortening the duration of the breeding process and reduce the cost of selection.

## Figures and Tables

**Figure 1 ijms-22-05423-f001:**
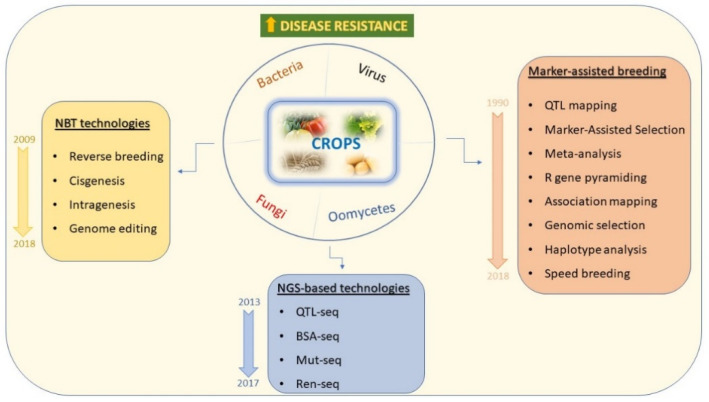
Schematic representation of genetics and “omics” approaches to study and solve disease resistance in crops, reviewed in this report (NBT: New Breeding Technologies; QTL: Quantitative Trait Locus; BSA: Bulk Segregant Analysis; Mut: Mutant; Ren: R gene enrichment; R: Resistance).

**Figure 2 ijms-22-05423-f002:**
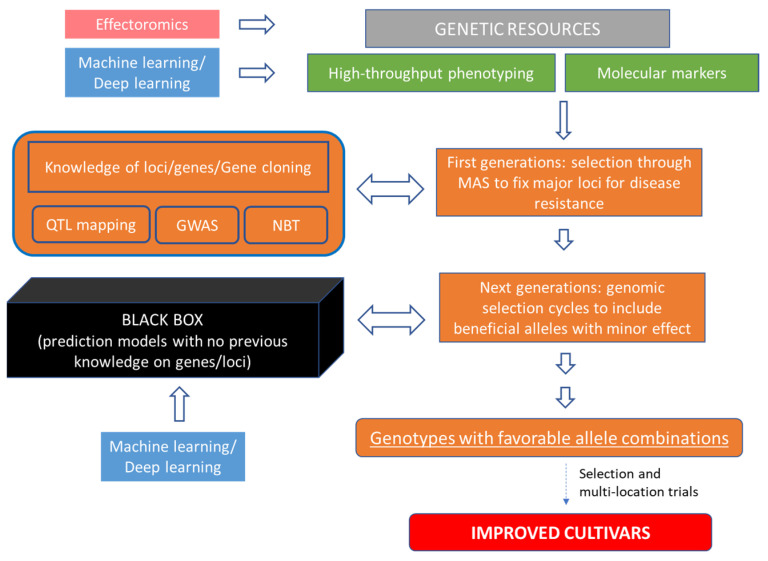
Workflow by integrating the different molecular approaches to obtain improved cultivars for disease resistance in crops.

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
