# Peer review of "Genomic Approaches to Identify Molecular Bases of Crop Resistance to Diseases and to Develop Future Breeding Strategies"

_ijms, 2021, doi:10.3390/ijms22115423_

Round 1

Reviewer 1 Report

The review article "Breeding for improvement of crop disease resistance: from markers to post-genomic approaches" is dedicated to the analysis of different approaches, from markers to recent genomic and "post-genomic era" technologies, necessary for better understanding of the complexity of host-pathogen interactions and genes,  including those with small phenotypic effects, and mechanisms that underlie resistance. The authors of the article believe that an efficient combination of these approaches can be the basis to develop an efficient breeding strategy to obtain resistant crop varients that yield higher in increasing disease scenarios. 

The article is well written.

The study has a good design.

The article is logically divided into sections and subsections.

In the article there are no grammatical and stylistic errors.

There is a table and figures of good quality presented in the article.

The references cited are relevant and adequate.

A large number of scientific literature sources were analyzed.

The work has a high degree of novelty.

In my opinion, this review paper can be recommended for publication after minor revision.

It is recommended to expand section "Effectoromics".

It is recommended to include a list of abbreviations, used in the article.

It is recommended to add articles of 2020-2021 to the list of references.

Reviewer 2 Report

Strengths

The review provides a comprehensive overview of the methods used in breeding for disease resistance in crops.  The written style is of good quality, and the extensive list of references cited are generally up to date.

Overall, the topic is a promising subject for review, though more justification and focus is needed

Weanesses

The article does not seem to be well-suited to the scope of the journal which is described as an ‘open access journal providing an advanced forum for biochemistry, molecular and cell biology, molecular biophysics, molecular medicine, and all aspects of molecular research in chemistry’. A journal more focussed on Agricuture would be more appropriate

The article lacks focus, and it is not clear why this particular topic is timely and appropriate for review. Consequently, the article comes across more as a collection of descriptive methods rather than focussing on a particular topic to make a point

The Introduction is long and rambling, and the content is repeated later on in the article. The Introduction should be limited to one or two paragraphs stating why the topic is timely and appropriate, building up to state the aims of the review and why it is needed.

Indeed, there are many reviews like this so it is not clear why this article is needed. For example, a quick search finds ‘Li W. et al Exploiting Broad-Spectrum Disease Resistance in Crops: From Molecular Dissection to Breeding. Annual Review of Plant Biology Vol. 71:575-603 (April 2020) https://doi.org/10.1146/annurev-arplant-010720-022215’ . This title is very similar to that of the proposed article, so a clear justification for it is needed.

Overall, whilst the article is quite comprehensive it is also quite dense. The individual topics are certainly relevant and topical, but the dense nature of the writing makes it difficult to follow. Within each sub-topic, the subjects tend not to be well-connected. Consequently, it is difficult for the reader to follow the thread of dialogue and understand the main points.

The illustrations are rather simplistic but could potentially be useful additions to the article. It might be helpful to pass these round colleagues to see how they could be improved

Effectoromics seems outside the scope of the main article and is not mentioned again in the overall conclusions and prospects

Round 2

Reviewer 2 Report

There is a slight improvement in the manuscript, and its purpose is more clearly defined. It is still quite a dense read, more like a list of techniques rather than forming a synthesis as a basis to direct the way forward. There is still an opportunity to improve this, perhaps stating more clearly the historical development of molecular techniques as mentioned in the reply letter. English in some of the modified sections should be improved.

It is understood that the 'special Issue' is appropriate for this article

Author Response

Dear reviewer,

We modified the Figure 1 which represents a scheme of the molecular and genomic approaches reviewed in the manuscript, ordered based on a temporal scale of their historical development as retrieved by literature. Some sentences have also been added throughout the text to clarify the historical relationships between different approaches.

The last modified sections have been improved for the English language.

Thanks for your comments. We hope now It's suitable for publication.

Best regards,

Daniela Marone

Round 3

Reviewer 2 Report

There have been some improvements made to the manuscript, and these have been sufficiently explained by the authors.